# Barriers and facilitators to physical activity among Black women: A qualitative systematic review and thematic synthesis

Sherron L. Howard[1]*, John B. Bartholomew[2]

1 Department of Kinesiology and Health Education, The University of Texas at Austin, Austin, Texas, United States of America, 2 Department of Kinesiology, The Pennsylvania State University, University Park, Pennsylvania, United States of America

* slhoward@utexas.edu

## Abstract

American Black women are less physically active than other American women. While a number of qualitative studies have investigated this issue, there has been no effort to synthesize this literature. *Purpose*: This study aimed to utilize qualitative thematic synthesis to employ the intersectionality framework in synthesizing existing published qualitative studies on barriers to and facilitators of physical activity for Black women and reflect constructs related to (a) all American women, (b) Black Americans, and (c) Black women. This qualitative thematic synthesis included 18 studies published from 2011 to 2023. Studies were coded for quality and then reviewed with their themes identified and inductively integrated across the studies. The resulting themes were then deductively placed within a broader structure of the intersectionality framework. Reflecting this framework, three categories were identified: (1) general experience of physical activity for women, regardless of race, with themes of (a) motivation, (b) structed support, (c) overall health, (d) environment; (2) general experiences of physical activity for Black Americans, with a sub-theme of (a) low access; and (3) specific, intersectional experiences of physical activity for Black women, with sub-themes of (a) black hair, and (b) body ideal. Our synthesis of the existing qualitative research revealed that Black women experience PA related to being a woman, to being Black, and to the intersectional nature of being a Black woman. Interventions might target these intersectional themes to tailor interventions to support PA in Black women.

## Introduction

Around 47 million people self-identify as Black, making up 14.2% of the United States [1] population with 51.8% being women [1]. Unfortunately, this population reports high levels of physical inactivity. As of 2020, non-Hispanic Black people have a 30% prevalence of inactivity outside of work [2]. Likewise, national surveillance data shows that only 27% to 52% of Black adults participate in regular physical activity, with Black women reporting lower levels of physical activity than Black men regardless of social determinants, e.g., socioeconomic status, and

**Funding:** The authors received no specific funding for this work.

**Competing interests:** The authors have declared that no competing interests exist.

education [3]. Black women also have higher rates of cardiovascular disease, high blood pressure, and type 2 diabetes [4]. This is not surprising, as the links between physical activity and health share well-documented, biological mechanisms [5, 6]. Despite a history of epidemiological research [7], we do not have a firm grasp on why Black women report lower levels of physical activity.

Potentially, contributors to behavioral choice derive in part from cultural factors [8] that might underlie differences between Black and other women. This has been most clearly studied for differences in body composition as White women are more likely to be motivated to engage in physical activity to achieve a thinner ideal than Black women [9]. It is also well-documented that Black communities view full-bodied women as attractive [10], with Black women reporting greater body satisfaction than White women [11]. This has been attributed to traditional African societies viewing women with heavier shapes as healthy and wealthy, with these attributes setting the standard for beauty [12]. Black women tend to prefer the "coke bottle" shape or "thick" body that comes with profound breasts, a narrow waist, round hips, and a prominent buttock [13]. As a result, women of African descent are less likely to conform to the Western beauty standard of thinness [14–17]. Black women are less fearful of gaining weight and more likely to promote physically heavier models as the ideal body size than White women [18]. As a result, Black women, whose body mass index would rate them as obese, have higher self-esteem, body satisfaction, and overall satisfaction with their appearance than similarly sized White women [19, 20]. Moreover, Black women report less body satisfaction when underweight than when overweight [18, 21]. While Black women who were socialized in a White social setting are more likely to prefer a thin ideal [22], they still report that the Black community and Black men prefer a heavier, physical shape–promoting acceptance of a larger body size [20, 22, 23]. While differences in cultural ideals for body image serve to improve the mental health of Black women, it may remove a key motivational factor that contributes to the discrepancy in levels of physical activity between White and Black women. Thus, rather than superimposing a White ideal to the experience of Black women, it is important to consider their unique experience of their bodies [24].

Crenshaw's work introduced essential terminology to describe the marginalization of Black women at the intersection of race, gender, and class. While Black women have long recognized these dynamics, now known as intersectionality, Crenshaw's framework helps academics better understand the overlapping systems of oppression they face. Rooted in Black feminist thought and critical legal studies, the concept highlights how Black women struggled to win legal cases based on either gender or racial discrimination alone, as both forms of discrimination coexist, shaping their experiences and oppression [25]. While developed in the legal field, the intersectionality framework has been widely applied to health issues [25]. However, it has not been applied to understand the experience of Black women regarding physical activity. This paper is designed to address this void.

Black women's experience includes overlapping marginalization, being Black and being a woman. As a result, and Black women broadly experience three categories of barriers related to physical activity. Those that are common to American women, those that are common to American Black people, and those unique barriers experienced by American Black women [26, 27]. This, in turn, creates unique impacts on physical activity for Black women that are distinct from the other groups [9]. Thus, the experience of Black women is not merely the additive effects of being Black and a woman, with Black women having to combat different forms of marginalization arising from more than one status [28]. Instead, the intersectionality of being Black and a woman raises unique experiences of discrimination and privilege beyond the common challenges they face with other women and other Black people. For example, conciseness surrounding hair is uniquely experienced by Black women, whose time and financial

investment is significantly greater than other groups of women [22]. Data from Donahoo (2023)'s study suggests that White beauty standards and anticipated employer expectations influence Black women's hairstyle choices. It is, therefore, surprising that the existing literature has failed to apply an intersectionality framework to understanding the physical activity behavior of Black women. This likely undermines efforts to understand barriers and facilitators to physical activity and to develop interventions to target Black women.

Black women in America also experience the residual effect of racial segregation. In *The Truly Disadvantaged* (1987), Wilson argued that urban poverty, particularly the development of a ghetto underclass, was the result of structural economic changes and the departure of middle- and working-class Black families from inner-city ghettos [29]. Benefiting from civil rights advancements like affirmative action and anti-discrimination laws, these families moved to better neighborhoods, leaving behind poor Black communities. This exodus removed a critical "social buffer," leading to increased joblessness, welfare dependence, and lawlessness in socially isolated areas lacking resources and job networks [29]. These residual effects add to the lack of access to green space and safe areas to be physically active [30]. As such, it is likely that some of the patterns of physical activity derive from these differences.

A recent narrative review by Obi et al. (2023) tried to understand physical activity and African women. Their findings identified significant barriers to physical activity such as body image perception, hair care concerns, gender norms, fear of sexual stereotypes, and family responsibilities that limit their involvement in physical activity. There were, however, several limitations to this paper. It drew studies from multiple countries. While this is a strength as it reflects broad differences across countries and barriers common to Black women of African descent, it does not reflect the specific cultural factors that might influence physical activity in Black women in America. Given the unique experience of Black people in America, it is not clear that the experience of Black women in other countries would generalize to the United States. In addition, they completed a narrative review that included a small set of qualitative studies along with mostly quantitative research. This excluded the majority of the existing qualitative research.

Barnett & Praetorius (2015) conducted a qualitative-interpretive meta-synthesis on African American women and their nutrition and physical activity. They found themes of family, structured support, translating knowledge into behavior modifications, barriers, and God is my healer. Their themes were intertwined one with another and only one theme could clearly be labeled physical activity related (barriers). This is a limitation of their study. An additional limitation that can also be considered a positive of their study is they focused on cultural influences on nutrition but not on physical activity. A broader thematic synthesis of the available qualitative research on American Black women is required to better understand the cultural experience of this population with regard to physical activity. Moreover, both studies lacked a theoretical framework to guide the interpretation of their data. This present review will address this through an application of the intersectionality framework.

A qualitative synthesis allows for a critical review of the existing, qualitative literature on a topic [31]. It applies a rigorous assessment of study quality while synthesizing the responses of participants across the existing research. In addition, core themes are re-assessed and interpreted across studies to provide a broader understanding of the data. In this case, our review includes responses from a large group of Black women, drawn from multiple qualitative studies. Their range of responses might reveal different categories of barriers and facilitators than is possible in a single study and small group of participants. Thus, the purpose of this study was to synthesize available published qualitative studies on physical activity barriers and facilitators for Black women using the intersectionality framework. The use of the intersectionality framework is intended to provide a lens to better understand the unique experiences of Black

women. By synthesizing existing qualitative literature, within an intersectionality framework, we hope to better understand the different forms of barriers and facilitators for physical activity as reported by American Black women. This, in turn, should help guide interventions that target Black women for increased physical activity.

## Methods

### Approach

Following the identification of studies that fit the inclusion criteria, we assessed the quality of each study utilizing the Critical Appraisal Skills Programme [32]. We then used an analytic approach that blended both inductive and deductive methods. Thematic analysis was employed inductively to classify these data and formulate novel themes that extend beyond the content of the initial study findings. Themes were then deductively grouped into components of the intersectionality framework.

### Instrumentation

It is important to self-identify the potential for bias that flows from the Authors' positionality as the author is the instrument in qualitative research.

### Positionality

The lead author is a Black woman with a family history of diabetes and high blood pressure. The women in her family have developed physical ailments that make it difficult for them to be physically active. She enjoys physical activity, utilizes it to maintain her mental health, and is pursuing a doctorate to understand motives for physical activity better. The second author is an older White, male with a similar family history of diabetes and high blood pressure. He enjoys physical activity and actively commutes to work. He holds a PhD in Exercise Science and teaches and engages in scholarship on physical activity and public health.

### Sample of literature

The search terms were: Black women, African American women, physical activity, exercise, qualitative, interview, and focus group, with the search from 2011–2023. Barnett & Praetorius (2015) completed a qualitative interpretative meta-synthesis on articles published before the year 2011, therefore we focused on articles that were published from the year 2011. Our final search was conducted on 11/13/2023; thus, all articles published from 2011 until the beginning of November are included in our current qualitative review. These words and years were used in the following databases: PubMed, Ovid Medline, PsychInfo, CINAL, and Web of Science. The inclusion criteria were: 1) published in peer-reviewed journals, 2) published in English, 3) included a sample of American Black women, 4) used qualitative or mixed methods design (e.g., interview, focus group), 5) explored physical activity barriers and/or facilitators for Black women, and 6) direct quotes about physical activity barriers and/or facilitators. Our search was limited to research conducted in the United States as we hypothesize cultural influences as well as Black Americans' experiences are different from Black people in other countries. The first author ran all database searches and screened each article based on the inclusion criteria. The search yielded 6,783 articles. After a title review, 6,611 articles were excluded leaving a total of 172 articles. After reviewing abstracts, we removed 146 articles for not having Black focus groups, duplicates, no physical activity themes, and not being a qualitative method. This left 26 articles. Seven of these were removed for not having quotes about physical activity. One was eliminated for including

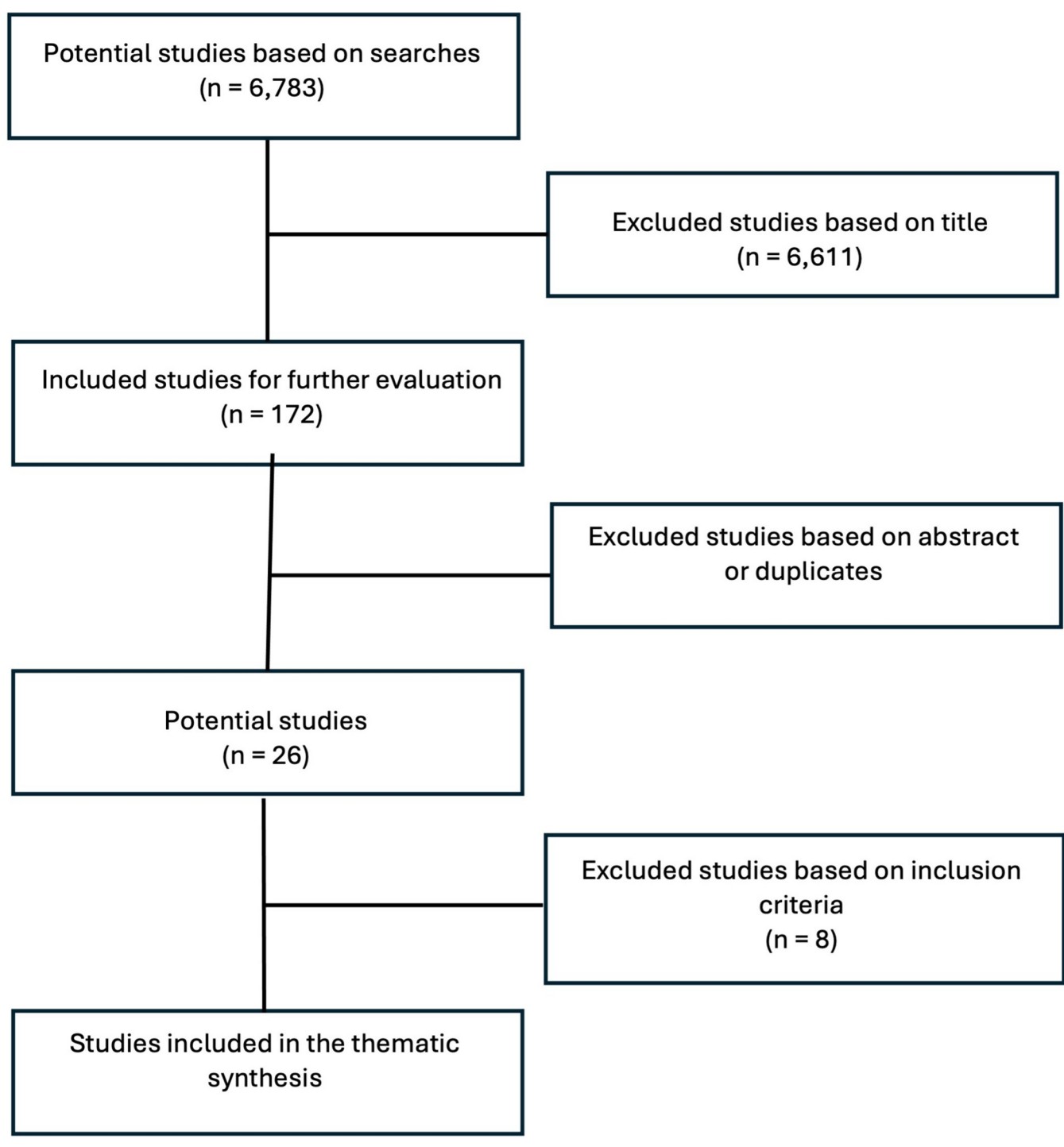

**Fig 1. Visual representation of the sampling process.**

both physical activity and nutrition without physical activity quotes. Fig 1 shows a graphical representation of the sample of literature steps. Table 1 provides the detailed demographics of the final 18 studies.

Table 1. Demographic data and themes extracted directly from the eighteen studies included in this review.

| Authors (year) | Study Objective | Themes as Coded by the Author | Data Collection Strategy | n | Age | Location | Quality Check Total Score |
|---|---|---|---|---|---|---|---|
| Baruth et al. 2014 | To describe barriers to exercise and healthy eating identified through focus groups with women recruited from urban neighborhoods of high poverty in the southeastern United States. | Personal Barriers<br>Social Barriers<br>Environmental Barriers<br>Race and Culture | Focus groups | 28 | 25–50 | Columbia, SC | 7.5 |
| Dlugonski et al. 2016 | To explore perceptions of physical activity, health status, and motives and barriers for physical activity among low-income, Black single mother | Health Behavior Motives<br>Health Behavior Barriers | Semi-structured focus groups | 32 | 18–53 | United States | 7.5 |
| Doldren & Webb 2013 | To explore the views of Black women in Florida to identify whether knowledge or attitudes to healthy food and physical activity were related to motivations for healthy behavior for working age Black women. | Knowledge and Motivation<br>The family: learning from, and being, a role model<br>Facilitators and barriers: preplanning and 'laziness' | focus groups | 40 | 18–45 | Broward County, Florida | 7 |
| Evans 2011 | To report the findings of a qualitative study of southern rural Black women's understanding of exercise and their personal exercise practices. | Global Factors<br>Thoughts about exercise and the heart<br>Exercise and the heart<br>Internal barriers<br>Physical barriers<br>Mental/Emotional barriers<br>External barriers<br>Other obligations<br>No partner<br>Environmental barriers<br>Reasons to exercise<br>Encouragement<br>Improves other conditions<br>Relieves stress/Makes me feel better<br>Taking action<br>Current practices<br>Need to exercise more<br>Preferred exercise<br>How much exercise? | Interviews | 20 | 40–60 | Southern rural county (AR) | 9.5 |
| Harley et al. 2014 | To examine the successful experiences of low-income African American women who had, over the long-term, integrated physical activity into their daily life. | Challenges to maintaining physical activity<br>Financial constraints<br>Physical strain<br>History of sedentary relapse | Interviews | 14 | 26–65 | Boston, MA | 9 |
| Hubbell et al. 2020 | To describe the perspectives of AA women between the ages of 35 and 65 years, who have been physically active long term. | Individual Level<br>Being healthy<br>Having increased health awareness<br>Managing their lifestyle to remain physically active<br>Social Level<br>Role modeling<br>Observing quality of life with friends<br>Family and coworkers<br>Establishing mutual support of PA<br>Environmental Level<br>Using the neighborhood as a venue to perform PA with friends and family<br>Using the work environment as a venue to encourage PA | Interviews | 14 | 35–65 | Midwestern US | 8.5 |

*(Continued)*

**Table 1.** (Continued)

| Authors (year) | Study Objective | Themes as Coded by the Author | Data Collection Strategy | n | Age | Location | Quality Check Total Score |
|---|---|---|---|---|---|---|---|
| Huebschmann et al. 2015 | To identify physical activity barriers and facilitators in AA women with a focus on whether and why sociocultural hairstyle maintenance factors influenced physical activity behavior. | Barriers:<br>Perspiration: The burden of "sweating my hair out"<br>Time burden of "sweating my hair out"<br>Financial burden of "sweating my hair out"<br>Sociocultural norm for hairstyle appearance<br>Environment—Drying effects on hairstyle | Focus groups | 52 | 19–73 | Metropolitan Detroit area | 9.5 |
| Joseph et al. 2017 | These qualitative assessments focused on: (1) collecting empirically-driven data regarding AA women's perceptions, manifestations, and determinants of PA to improve clarity and cultural relevance of our PA program; and (2) enhancing the theoretical fidelity of the 5 SCT constructs among AA women targeted by the PA intervention–Self-efficacy, Self-regulation, Behavioral Capability, Outcome Expectations, and Social Support. | Behavioral Capability<br>Lack of knowledge on the US PA Guidelines for health<br>Ambiguity in the terms "exercise" and PA<br>Desire for a specific exercise prescription and plan<br>When initiating a PA program, a strong social support network is needed<br>Outcome Expectations<br>Improved health outcomes, including: Reduced risk for high blood pressure and type 2 diabetes Reduced stress Improved sleep<br>Weight loss<br>Increased energy and ability to engage in more activities with their family<br>Being a good role model to others<br>Self-efficacy<br>PA was encouraged and enjoyed as children through games and sports<br>Negative experiences with PA during childhood were associated with forced sporting activities or concerns about body development<br>Negative experience with PA as adults were related to engaging in PA that was too intense or beyond their skill level<br>Social Support<br>Family members emerged as a key source of social support<br>Discouragement from family/friends for PA was negligible<br>Messages of social support should be positive and encouraging; criticism should be avoided as it is not viewed as motivational<br>Women desired to hear personal testimonials from other AA women on how they have successfully increased PA<br>Self-regulation<br>Women identified a variety of strategies to incorporate more PA into their daily schedules<br>Scheduling PA into daily schedules was preferred<br>Women reported previous attempts with self-monitoring, but ultimately stopped due to burden or loss of interest<br>Women emphasized the importance of setting realistic short-term goals to achieve long-term goals<br>Potential reinforcements/rewards for meeting PA goals included monetary incentives, clothing, and words of encouragement/praise. | Focus groups | 25 | 24–49 | Greater Phoenix area | 8.5 |

(*Continued*)

**Table 1.** (Continued)

| Authors (year) | Study Objective | Themes as Coded by the Author | Data Collection Strategy | n | Age | Location | Quality Check Total Score |
|---|---|---|---|---|---|---|---|
| Joseph et al. 2018 | To qualitatively explore how hairstyle preferences and hair care maintenance practices of AA women can limit PA engagement, uncover the underlying reasons of why hair is viewed as a barrier to PA among some AA women, and provide intervention design strategies for researchers and practitioners to leverage in the development of culturally targeted PA programs for AA women. | Impact of perspiration on hair and hairstyle maintenance<br>Image and social comparison<br>Solutions to overcome hair-related barriers to PA | Focus groups | 25 | 24–49 | Greater Phoenix area | 9 |
| Justin & Jette 2021 | To center participants' desire for "thickness," and the relation of that desire to their understanding of Black femininity, as a critical element to how they navigate obesity and public health discourse. | Questioning the "dominant obesity discourse": "That chart ain't for us, that's for them"<br>Reifying biological determinism: "There is something coded in our DNA"<br>"Thick" body politics and black femininity: "It's something that's engrained" | Interviews | 8 | 18+ | Maryland | 9 |
| Krans & Chang 2011 | A qualitative description of pregnant African American women's: (1) barriers to exercise during pregnancy and (2) facilitators that could increase the amount exercise performed by pregnant African American women. | <u>Individual/Intrapersonal Barriers</u><br>Physical Limitations<br>Lack of Time<br><u>Information Barriers</u><br>Lack of Healthcare Provider Guidance<br><u>Resource Barriers</u><br>Financial Constraints<br>Lack of Neighborhood Facilities and Resources<br><u>Socio-cultural Barriers</u><br>African American Cultural Influences | Focus groups | 34 | 18–30 | United States | 9 |
| Mama et al. 2015 | To gain community insight into individual, social, and environmental factors that influence physical activity adoption and maintenance in African American and Hispanic women using the ecological model as a guiding framework, which accounts for interactions among individuals and their social and physical environments. | Intrapersonal Factors<br>Interpersonal Relationships<br>Accessibility and Safety<br>Caretaking for Others | Interviews | 11 | - | Austin & Houston, TX | 9 |
| Miller & Marolen 2012 | To explore physical activity-related experiences, perceptions, and counseling expectations among urban, underactive, African American women with type 2 diabetes. | General physical activity/exercise perceptions<br>Barriers to physical activity<br>Enablers to physical activity<br>Comparison of physical activity counseling expectations and experiences<br>Physical activity-related health responsibility<br>Greater concern for health care-related needs of others | Focus groups | 11 | 21–50 | Nashville, TN | 9 |
| Price et al. 2013 | Among older Black women who have demonstrated a commitment to regular PA, what led them to initiate PA and how have they successfully maintained PA? | PA Initiation<br>Integration of PA in Daily Life<br>Temporary Lapses<br>Strategies for Maintaining PA<br>Regular Routine<br>Flexible Planning and Routine Adaptation | Interviews | 15 | 60+ | Eastern coast of US | 7.5 |

(*Continued*)

**Table 1.** (Continued)

| Authors (year) | Study Objective | Themes as Coded by the Author | Data Collection Strategy | n | Age | Location | Quality Check Total Score |
|---|---|---|---|---|---|---|---|
| Redmond et al. 2022 | Looked at barriers and facilitators of physical activity and maintain healthy eating habits among younger African American women to better understand what motivates and hinders older African American women from participating in these healthy behaviors. | Barriers<br>Interpersonal issues<br>Subtheme for physical activity barriers: Pain, time, motivation to be active<br>Limited definition of physical activity<br>Facilitators<br>Routine of regular physical activity<br>Influence of family | Semi-structured interviews | 26 | 55+ | Kansas and Missouri | 8 |
| Sebastião et al. 2014 | Explored the understanding of physical activity among inactive older African American women. Moreover, facilitators of and barriers to being physically active in their local environment were also examined. | Social Facilitators<br>Environmental Facilitators<br>Personal Barriers<br>Environmental Barriers | Interviews | 7 | 65–75 | Urbana-Champaign, IL | 9 |
| Sebastião et al. 2015 | Employed a mixed-method approach to explore perceptions of PA among Regularly Active and In-Active older African American Women and the factors that positively or negatively contribute to their decision to be physically active. | Knowledge and perceptions of physical activity<br>Factors influencing physical activity participation<br>Physical health and psychological and emotional health<br>Financial problems<br>Family demands<br>Unsafe neighborhood | Focus groups | 20 | 60–80 | Central Illinois | 9 |
| Sweeney et al. 2019 | To evaluate 1) the different sources of autonomous versus controlled motivation that drive African American women's PA engagement; and 2) the extent to which differences in autonomous motivation relate to different PA needs and interests at individual, interpersonal, and environmental levels that could be used to inform intervention development. | Past PA Engagement<br>Sources of Motivation for PA<br>PA Barriers<br>Resources needed for PA<br>Role of family and friends | Focus groups | 31 | 24–69 | Suburban Southeastern Community | 9 |

## Quality check

The Critical Appraisal Skills Programme is used to assess the validity, relevance, and applicability of qualitative research studies [30]. Table 1 shows the total score given for each study. Studies received a score of 0 (no), 0.5 (can't tell), or 1 (yes) in ten categories: (1) clear study aim, (2) methodology, (3) appropriate design, (4) recruitment strategy, (5) data collection, (6) bias, (7) ethical considerations, (8) data analysis, (9) clear findings, and (10) value of research [30]. Out of a possible high score of 10, 14 of the 18 studies received a score of 8 or above with one study receiving 7 and three studies receiving a 7.5. The common limitation centered on bias in recruitment. Thirteen of the 18 studies received a low score of 0 for bias and the remaining five received a score of 0.5. This reflected our interpretation that bias was not considered when creating interview questions and during analyses. The other aspects of the quality review were more consistently strong across studies. As a result, no studies were eliminated due to low quality.

### Data extraction

This step was to extract the themes from the original studies, which would serve as the data for analysis. The themes from all 18 studies were extracted by the first author, as written, verbatim by the author to maintain the integrity of each study. These are shared in Table 1.

### Data analysis

The analysis was designed to synthesize and translate the themes to reflect the shared understanding of Black women and their intersectional barriers to physical activity across the studies. First, both authors synthesized and translated the original, source themes into new overarching themes that cross studies. The process of synthesis entailed evaluating the similarity between the extracted themes and consolidating them into initial categories for translation. After synthesizing the themes, we proceeded with the translation, which resulted in a fresh set of themes and a more cohesive comprehension of the data. To achieve the appropriate translation of sources, a total of 18 separate studies that shared the experiences of 413 Black women whose ages ranged from 18 to 80 were incorporated (refer to Table 1 for demographic details). Translation was accomplished through collaborative efforts among the authors, involving theme development and result formulation. This involved a review of original quotes when available and, in some cases, movement of source themes to other categories of broad themes.

## Results

The purpose of our analysis was to review and synthesize available qualitative studies on physical activity in Black women by using the intersectionality framework to reveal barriers or facilitators specific to Black women. Our deductive analysis sought to separate these, intersectional barriers from other categories of barriers. To this end, our deductive, categories included: general experiences for women, race-related experiences of Black people, and intersectional experiences of Black women. Five themes emerged from our inductive analysis: (1) motivation, (2) structured support, (3) overall health, (4) environmental factors, and (5) intersectional experiences. Lastly, there were ten sub-themes: (a) habit, (b) intent to engage, (c) social norms, (d) social support, (e) health outcomes, (f) health literacy, (g) environment, (h) low access, (i) black hair, and (j) body ideal. The sub-themes serve as either facilitating factors to increase physical activity or barriers to reduce physical activity. We present these facilitators and barriers together to illustrate the variability amongst Black women in their experience of each theme. Table 2 shows how the original themes were recategorized into new categories, overarching themes, and sub-themes. The interpretive discussion of the disentanglement of these themes follows with quotes provided when available from the original manuscripts to illustrate the analysis.

### General physical activity experiences of women

These themes reflected a mix of barriers and facilitators that are common experiences of many women.

Motivation was viewed as a broad driver of physical activity that could serve as both a facilitator and a barrier–depending on where they sat on the continuum. Sub-themes included habit and intent to engage.

### Habit

Participants discussed how getting into a routine motivated them to engage in physical activity daily. A participant stated, *"The more I went to the activities the more I started to get up and pop*

**Table 2. Translation of themes.**

| New Overarching Theme | Sub-Theme | Authors |
|---|---|---|
| General Experiences of Women | Motivation<br>Habit<br>Intent to Engage<br>Structured Support<br>Social Norms<br>Social Support<br>Overall Health<br>Health Outcomes<br>Health Literacy<br>Environmental Factors<br>Environment | Evans 2011<br>Joseph et al. 2017<br>Price et al. 2013<br>Redmond et al. 2022<br>Sebastião et al. 2014<br>Doldren & Webb 2013<br>Evans 2011<br>Harley et al. 2014<br>Joseph et al. 2017<br>Mama et al. 2015<br>Miller & Merolen 2012<br>Redmond et al. 2022<br>Sebastião et al. 2014 Sebastião et al. 2015<br>Sweeney et al. 2019<br>Hubbell et al. 2020<br>Joseph et al. 2017<br>Redmond et al. 2022<br>Sebastião et al. 2014<br>Evans 2011<br>Joseph et al. 2017<br>Krans & Chang 2011<br>Mama et al. 2015<br>Miller & Merolen 2012<br>Hubbell et al. 2020<br>Joseph et al. 2017<br>Sebastião et al. 2014<br>Evans 2011<br>Joseph et al. 2017<br>Redmond et al. 2022<br>Hubbell et al. 2020<br>Sebastião et al. 2014 |
| Race-related Experiences of Black People | Low Access | Evans 2011<br>Dlugonski et al. 2016<br>Krans & Chang 2011<br>Mama et al. 2015<br>Redmond et al. 2022 |
| Intersectional Experiences of Black Women | Black Hair<br>Body Ideal | Huebschmann et al. 2015<br>Joseph et al. 2018<br>Price et al. 2013<br>Baruth et al. 2014<br>Justin & Jette 2021 |

*up out of the bed in the morning, and I thought I, you know, just kind of hit me why. I wake up in the morning and I'm not, you know, stiff or I'm not—I don't know how you could describe that. It just all of a sudden, I realized, wow, I'm not stiff anymore." [33]* An additional participant discussed the importance of planning in advance to create the habit of being physically active. *"It's not something I do now, but in the past, one of the biggest things was just planning and preparation. . . if I had a plan in advance, it's a lot easier to follow through and to stay on top of it" [34].*

## Intent to engage

However, most participants expressed a general lack of motivation to perform physical activity [35]. For example, one participant compared their lack of motivation to smoking, *"I really, really, really do want to be. I just lack motivation, so when I think about sometimes you meet someone and they say, "Well, I know I need to stop smoking, I got cancer, but I need to stop*

*smoking"–I understand now, so I empathize with them because it's not that you don't care—it's just something that you're not changing [36].* In contrast, other participants emphasized a negative motivation for PA, *"... There is some type of apprehension in the back of my mind, and I'm trying to figure out why, but I really need to say, "Go ahead, start doing it." I guess I feel that if I start, I'm going to have to continue. It's going to change my routine. [And that] Moves me out of my comfort zone." [36]* Many participants expressed a lack of motivation and/or confidence to do physical activity [36] despite a general desire to perform physical activity, but the motivation was lacking to carry that desire to action [37].

Structured support refers to the different areas of support needed to engage, maintain, or disengage from physical activity with two sub-themes of social norms and social support.

## Social norms

Some women discussed including their family in their physical activity to help with accountability. *"It, it helps me because, like I said, a lot of times when I get off work... I don't feel like doing anything... so what they [family] have done for me recently is ... [they] still hold me accountable; 'let's go walk outside' or... if we're going to the mall, 'let's just take two laps,' even though we're going to the mall for something else"* [38]. Others utilize their family and want to remain involved as a motivator for physical activity. *"For me, I think it comes to normalcy, I guess. So, normalcy [referring to the benefits of being active] to be in a category with the rest of my peers, my family, and being able to maybe keep up with family–with my son and my husband–and being able to do those things and not be exhausted or not sit out the ride or sit out the trip for whatever reason." [38]* Another expressed how physical activity allowed her to socialize with her friends and adapt to different kinds of physical activity. *"[Be]cause we all do very different kinds of physical activities... it made me... adapt in what I felt like I was willing to... to try to be able to do. And now I've found some new things that I really enjoy." [38]* An additional participant agrees that having active friends motivates them to continue and allows them to be held accountable. *"I think having a friend who is at your level or maybe somewhere around your level as far as physical activity, that's definitely an encouragement and you can both hold each other accountable and things like that." [34]*

## Social support

Many participants discussed the importance of social support when it comes to physical activity. *"I think I'm a good influence... I'm kind of a self-motivated person, but a lot of people want me to motivate them, so I have to call them. "Let's go out and walk." Nobody calls me. I'm the one. So maybe if I get somebody to tell me that also I'd go out more often."[36]* Similarly, participants agreed that they prioritize other's health over their own and would do so when it comes to increasing their physical activity [33]. *"If a family member or friend came up to me and they were sick and said, 'Honey, if I don't exercise, then I'm going to die. I would exercise with that person if it kills me. I'll do it for somebody else quicker than I'll do it for me" [35].*

Participants also discussed the importance of their social environment and the potential motivating force of seeing their neighbors being active [36]. In addition, some quotes were interpreted in the original paper as particular to the Black experience, but our analysis indicated broader, issues with social support. For example, *"It's not promoted enough in... African Americans. And I think it's from your family. If you didn't come from a family that exercised and I'm going to tell the truth, shame the devil, my parents didn't exercise. My grandparents didn't exercise." [39]* Many participants mentioned that their family members did not exercise regularly nor talked about the importance of exercise [39]. While the participants are speaking from their experience as Black Americans, and the original manuscript placed this in a racial

framework, our view is that multi-generational modeling is a form of social support for people of all races and ethnicities and should be included in the general experiences of all people.

Overall health refers to the physical, mental, and educational health of all people with two sub-themes of health outcomes and health literacy.

**(e) Health outcomes.** Participants note how important physical activity is for aging, energy levels, and mental health. *"I started getting the results I wanted, [and] it became less about that and more about feeling good. Being able to do a workout provides me with. . . a mental and physical stress release."* [36] Another participant added, *"I just want to keep this up, and I just feel like it's necessary as I continue to age, and I'm a single woman. . .. I want to be able to take care of myself for a long time (laughs)."* [38] Other participants emphasize the importance of physical activity to live a long healthy life. *"For me, the motivating factor is I want to live. You know. Straight up, I want to live. . .you see people who are passing away at your age, I guess. And, you know, because of heart attacks or, you know, things, diabetes or whatever these things, all these health obstacles and just different challenges because of, I'll say laziness, just being real about it, you know."* [34] The desire to live a long healthy life with the added benefit of stress relief helps them to engage in physical activity.

**(f) Health literacy.** Participants described confusion with the recommendations for daily physical activity. *". . . Not thought-out, but some people might not know what moderate is, you know, to intensity. . . because you do have a lot of what you call the average Joe, they're not middle class, they're not gone to college, half of them really basically have not even finished high school. . . people like that, which is call the average Joe, you are going to have to do–break it down in just a little bit more, what they said, layman's terms. . ."* [40] This participant is emphasizing the importance of presenting information at a level that is more accessible.

## Environmental factors

This theme housed a sub-theme that was related to all people (environment) and one that was specific to Black people (low access).

## Environment

Participants discuss the importance of green spaces and how that is a facilitator to be physically active. *". . .and this is like in the back of my house. . . you know there is a big field and there is like a walkway through the park district. . . they put walkways and bikeways. I just like the nature, it just lifts you up, it is just beauty. It is a beauty of nature and it makes you. . .eyes get in tune with nature. I love to hear the birds, I love to hear the crickets, I love all of that; the creatures that are on the earth, now does that make sense?"* [40] Participants also discussed the importance of neighborhoods and workplaces and how these places influence their physical activity. *". . . I took on a. . . position at an insurance company. . . that job required me. . . [to] read death certificates all day, as well as medical records. So, I was spending a lot of time. . . reading about the things that are killing us [African Americans] off. . . that was really a catalyst. . ."* [38] Another participant mentions, *"I get an opportunity to educate patients a lot. . . so even if they're not diabetic, or even if they do not have high blood pressure. . .. you can educate [about exercising]."* [38] This participant utilizes their environment to promote the importance of physical activity.

## Race-related experiences of Black people

There was one sub-theme that reflected the experience common to Black men and women as they relate to the historic marginalization of this population.

## Low access

Numerous studies described themes related to being Black and living in a historically marginalized neighborhood. Participants felt there were not enough recreational spaces–specifically in Black neighborhoods [39]. *"I think that the division between somebody being African American and Caucasian is like in a predominantly black neighborhood, there aren't facilities. . .but where a Caucasian person might live. . .they probably have more, you know, facilities where they can go to exercise"* [39]. Likewise, accessibility and safety were a common concern among participants in a second study [36] with these factors undermining their confidence that they could safely walk outside of the home. Finally, participants in a third study felt they had limited financial resources to pursue exercise [41]. *"I got so much other stresses on my mind. You're not worried about walking down to begin a physical activity program if you worried about paying your rent. I worry about using my money to pay my bills. I'm not worried about nothing else"* [41].

## Intersectional experiences of Black women

Finally, two primary sub-themes specifically reflected the experience of being a Black woman.

## Black hair

Two articles had a particularly strong response to the issues of styling and maintaining Black hair following PA as restyling hair was more time-consuming and expensive than for women of other races while a third mentioned hair [33, 42, 43]. *"I am going to a luncheon, I probably am not going to exercise that morning"* [42]. In comparison, *"For a corporate person, Black woman, it's like a daily issue [referring to 'sweating out' her hairstyle]. . .it's the whole image thing"* [42]. Interestingly, this applied to Black women of all ages, *"As a teacher during recess time, I noticed that the little black girls were not out on the playground playing, and they did not want to go in the gym, you know bounce the ball and do all of that because they were wearing braids and they had long hair and they said 'no mama said don't go out there and play because I will mess up my hair"* [42]. In addition to the barrier of time to style hair, they reported an added financial burden due to the increase in hair salon visits and purchases of more hair products as regular exercise increased the frequency of hair maintenance [42]. *"I've had friends that would stop workout programs if it meant that they were about to have a hair appointment. . .or wait until they [have] micro braids or in curls or whatever"* [43]. Additionally, participants noted the rules of some facilities did not allow the use of headscarves as a way to maintain their hairstyles. *"Some gyms won't allow you to wear hair scarves . . ., if you ever read their rules, they won't let people wear scarves on their heads"* [43].

   Participants also highlighted the differences between themselves and their counterparts when it came to lunch break physical activity and their hairstyles. *"I see my Hispanic and White counterparts; they're going on lunch break to exercise. I'm not doing that because I'm not going to look the same when I get back"* [43]. Participants felt their counterparts have different experiences when it comes to their hair and wished they would be respected based on their experiences. *"I feel like if White people really understood how our hair works and the things we have to go through, then they would just let it go. Because I don't 'necessarily have to wash my hair when I come home from the gym, but they'll be like' Eww, you didn't wash your hair"* [43]. Additionally, *"They're [referring to her coworkers] like, 'Oh, what did you do different?' 'Nothing. It's the same thing.' . . . But it's extra for us. We have to justify every hairstyle we have"* [43]. This lived experience can have effects on physical activity. *"If somebody came up with an answer to that hair issue, I think you would find that you could really get to them. I've said to some people that I've talked to that I know they are getting to that point where they're morbidly [obese] and I*

*say, "You mean to tell me that you would rather your hair be okay than to die?" They say "You've got to die from something." I mean that's what I've heard"* [33].

## Body ideal

Women discussed the cultural ideal of a Black woman's body, and that physical activity would undermine their efforts to be sexually attractive. For example, women expressed how exercising could cause them to reduce the size of their chest or hips and they did not want to be "stick thin". *"As a black woman you don't want to be seen without curves. You want to keep your curves"* [44]. Another participant explained a conversation with her doctor who did not share her view of the ideal body, *"Like she thinks I need to lose weight. And I keep explaining to her that a lot of my weight is in my hips and thighs and butt. Like what am I supposed to do about that? I can't be that way. I have muscle"* [45]. Other participants also equated "thickness" with attractiveness and muscularity and did not feel they were obese. *"Thick is like ok \*laughs\*. Fat is like there's some rolls. Thick is like they got some muscles they got a little booty. Hips but it's kind of toned in a way. That's thick"* [45].

The interpretation of the themes related to the intersectional framework suggests that there is a clear message that physical activity is focused on the experience of White women. While much of the existing qualitative literature on Black women seems to be oriented to reflect this focus, a distinct message is likely required to build on the strengths of Black women to promote physical activity. This interpretation is supported by a series of focus groups with active and inactive Black women [42]. Sebastião et al. (2015) showed that Black women have trouble understanding the Center for Disease Control guidelines and when presented with a revised document they still felt the requirements and suggestions were not representative of them. One active woman felt, *". . .for black women. . . there were restrictions on the physical activity for Black Americans. . . You couldn't even go swimming in the pool. You were restricted. You certainly couldn't hang out at the track in the high school and walk around the track at a high school and be suspect. You couldn't go to a public beach and swim. . ."* [46]. An inactive woman had similar sentiments, *". . . it has a lot to do with economics. . . looking at these lists [examples of activities provided in the materials for moderate and vigorous intensity] to me I still keep going back to the black versus white, whereas these are not the things we have thought of all of our entire lives"* [46]. This highlights the importance of representative information when promoting physical activity, as well as, considering the lived experiences of Black women when creating interventions.

## Discussion

This paper was designed to apply an intersectional framework to reflect the experiences of Black women when incorporating physical activity into their lives. This was accomplished by synthesizing the findings of 18 qualitative studies. Three categories were deductively used to identify intersectional experiences and differentiate them from other barriers, i.e.: general experiences of women, race-related experiences of Black people, and intersectional experiences that are unique to Black women. The inductive analysis of the 18 articles revealed five themes that reflected both barriers and facilitators: motivation, structured support, overall health, environmental factors, black hair, and body ideal. Motivation, knowledge, and confidence were expressed across the majority of the studies [33–37, 40, 46–50]. This is not surprising as these represent some of the most studied constructs in the history of physical activity research. They are common contributing factors and should be addressed when developing interventions for physical activity regardless of the gender, race, or ethnicity of the population. That is, Black women experience similar barriers and facilitators as most people who seek to adopt an

active lifestyle. Additionally, we found themes that centered on Blackness and the impact of historical poverty on knowledge and access to physical activity [44]. We felt these themes were race-related rather than intersectional barriers as they apply to both Black men and women and would be important to consider when designing interventions for Black people and other populations that have historically experienced marginalization due to racial inequality.

In contrast, only 4 of the 18 studies identified barriers that were unique to Black women. Two of these were specifically focused on the challenges of Black hair and the cost and time required to style hair following exercise [42, 43]. An additional study acknowledged hair could be a barrier for Black women without explicitly listing it as a barrier or theme [33]. Hair concerns were found to be passed along through generations, with one participant discussing the lack of activity for Black girls at recess due to concerns that it would impact their hair [42]. The other two studies focused on the perceived incompatibility of physical activity with the preferred body image for Black women [44, 45]. For many, physical activity is motivated by the thin ideal dominant in White culture [24]. In contrast, Black women seek a curvy, "thick" body type and express concern that too much physical activity will reduce these curves and make them less attractive. Body image has been consistently cited as a motivator for physical activity [45], but this has largely followed the White ideal of thinness [9] rather than reflect the ideal body image of Black women. For example, only one study mentions weight loss or body size as a motivator for physical activity [34] which is in direct opposition to the majority of work on White women. Previous reviews have also shown body image perception, hair care concerns, and family responsibilities [51] as a benefit and barrier to physical activity. Additionally, Barnett & Praetorius (2015) applied a cultural lens and identified race-related barriers. In contrast, the present review applied the intersectionality framework to disentangle motivators and barriers specific to Black women. This review also provides a better understanding of facilitators and barriers to Black women specifically. Therefore, it is critical to consider physical activity from the perspective of Black women and their unique considerations of physical attractiveness. Core issues with body image in Black women center on skin tone, hair, and body shape [22]. Given the importance of these factors, it is surprising that none of the papers mentioned skin tone and only three mentioned hair. Clearly, there is a need for further research to test the association between body image and physical activity for Black women, along with how to incorporate this information into interventions to promote physical activity in Black women.

It was interesting that participants in one paper discussed the potential to be motivated to be physically active if it benefited someone else's health [35]. This comment may reflect the Superwoman Schema framework. The Superwoman Schema refers to a set of beliefs and behaviors characterized by an excessive need to be successful, competent, and to excel in multiple roles, often at the expense of one's well-being [52]. This framework reflects a role orientation that was adopted to confront and transcend the challenging historical and sociopolitical context of gendered racism [52]. Of note, Black women often feel the need to help others before helping themselves, which may provide an avenue to intervene to build motivation. Behaving for the benefit of others also reflects more general theories of health behavior. For example, the Transtheoretical Model hypothesizes Environmental Re-evaluation as a key process of change where the individual considers the impact of their behavior on others [53]. Thus, it is not clear if this desire to be physically active to support others is common across people or if it is experienced to a greater extent by Black women. This is likely a useful area for future research.

Five of the included studies acknowledge race-related barriers, and only 4 of the 18 articles include any themes that were specific to the experience of the intersectional nature of being a

Black woman. This was a surprising pattern since the articles were specifically designed for Black women. This may be due to differences in the underlying theories that drove the structure and framing of their qualitative design. Two articles were specifically focused on the challenges of Black hair [43, 44] and were the only studies to report on these themes–despite these themes being supported in other reviews [8, 51]. Another [45] began with questions informed by Black Feminist Theory, which offers a critical lens to analyze and challenge the interconnected systems of racism, sexism, and other forms of oppression [54]. This resulted in a unique set of themes centered on body ideals. Of the remaining studies, only four reported themes specific to the intersectional experiences of Black women. Whereas the others took a more open and general line of questioning without a specific consideration or inclusion of an overarching theory or framework of race. It may be that more specific questions are required to create trust and gain insight into the experience of minoritized people, particularly when the area under study reflects the dominant White culture.

Qualitative research provides space for people to share their experiences unencumbered by the limits of a narrow set of items on a questionnaire [55]. Of course, people can only respond to the questions they are asked. For example, "Tell me about your biggest challenges to increasing your daily activity or sticking to a regular routine" [40]. This question may not elicit the response expected, but the use of the following "probe: how does your hairstyle influence your PA?" [43] helps to get at a response specific to the interviewee. Therefore, it is important to (1) acknowledge the researcher's positionality and (2) utilize critical theories tied to the positionality of the participants. That is, if we want to understand participation in physical activity for Black women, we might approach this research informed by the existing research on the unique experiences of Black women. One way to achieve this goal would be to utilize sociological perspectives when developing and implementing physical activity interventions for Black women. As is the case with every intersectional group, Black women are multifaceted, and their experiences are unique. For example, in one study the authors directly reflected the participants use of the term "laziness" as a theme [37]. This serves to perpetuate racial stereotypes of Black people when a more nuanced discussion of barriers to physical activity specific to Black people could have occurred. That is why the use of critical race theories (e.g., Black Feminist Theory, Superwoman Schema, Intersectionality Framework) helps to gain a deeper understanding of their perspectives of the world that are likely to inform their experience with physical activity.

### Limitations

A primary limitation is that we could only include 18 studies that fit the inclusion criteria. While this included 413 women across at least 12 states in the USA, it falls short of reflecting the population of Black American women, which should be taken into consideration before generalizing these results. Since only 12 states were included in the 18 studies, location could play a role in our findings since intersectional experiences can differ by region of the United States [56]. Additionally, there is a large age spread in the sample and barriers and facilitators can be experienced differently across generations. Future work should seek to systematically compare women of different ages. Finally, we focused only on Black American women to the exclusion of other cultures. This not only excludes women from other countries, but it also likely fails to represent the range of cultures within the American Black community [57].The intersectionality framework would suggest that physical activity barriers and facilitators will be experienced differently across cultures and this, again, requires additional study. Future research should also look at differences in views of physical activity by location as well as by age group.

## Conclusion

Despite these limitations, there are clear strengths. This is the first study to synthesize the existing qualitative research on physical activity for Black American women and to do so within an intersectionality framework. As a result, it provided a number of novel findings and highlighted the benefit of a sociological approach to the study of disparities in physical activity. As such, these results not only inform future research on health disparities but can be used to guide interventions that are specifically tailored to Black American women.

## Supporting information

**S1 Checklist. PRISMA checklist.**
(DOCX)

**S2 Checklist. PRISMA abstract checklist.**
(DOCX)

**S3 Checklist. CASP checklist.**
(XLSX)

**S1 Data.**
(XLSX)

## Acknowledgments

The authors thank Christy L. Erving, PhD, Earl W. Huff, Jr., PhD, Taylor Payne, PhD, and Jaylen I. Wright, PhD for their comments on early drafts of this manuscript.

## Author Contributions

**Conceptualization:** Sherron L. Howard, John B. Bartholomew.

**Formal analysis:** Sherron L. Howard.

**Methodology:** Sherron L. Howard, John B. Bartholomew.

**Supervision:** John B. Bartholomew.

**Writing – original draft:** Sherron L. Howard.

**Writing – review & editing:** Sherron L. Howard, John B. Bartholomew.

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
