## [Decision Letter · Decision Letter 0]

25 Jul 2024

PGPH-D-24-00822

Barriers and facilitators to physical activity among Black women: a qualitative systematic review and thematic synthesis

Dear Dr. Howard,

Thank you for submitting your manuscript to PLOS Global Public Health. After careful consideration, we feel that it has merit but does not fully meet PLOS Global Public Health’s publication criteria as it currently stands. Therefore, we invite you to submit a revised version of the manuscript that addresses the points raised during the review process.

We look forward to receiving your revised manuscript.

Kind regards,

Tsitsi B. Masvawure, Ph.D.

Academic Editor

Additional Editor Comments (if provided):

Thank you for your submission. As you will see from the reviewers' comments, your paper has several strengths. The main one is the focus on physical activity among black women in the USA. This is an important topic in public health. Additionally, some of the review findings highlight important factors that seem unique to black women, such as concerns with hair-care and the preference for 'thicker' bodies. Despite this, the paper has one glaring weakness, which was highlighted by one reviewer and which I would like to reiterate. The authors do not define what they mean by the term 'intersectionality' and it is not clear how this term is being used in the paper. In the results section, the authors state that the purpose of their analysis was to 'determine if there are intersectional themes...". They go on to note that their "deductive, intersectional categories included.....intersectional experiences of Black women.' It seems to me that the authors are using the term in two ways: first, to in the sense of 'overlapping' themes and the second, should use is more in line with Crenshaw's concept as refers to the the ways in which multiple identities come together to produce particular outcomes. However, the paper does not quite show how black women's physical activity is intersectional, or which parts are intersectional. It almost seems like the authors assume intersectionality (used in the Crenshaw sense) but do not make an effort to explain to the reader how things are intersectional. For instance, their section on intersectionality in the results focuses on hair and body image, but it is not clear why these two factors in particular have been singled out as 'intersectional' and the other factors reported in the paper have not. As one reviewer pointed out, most of the factors reported in the paper could be considered as speaking to intersectionality. The discussion further reflects the lack of clarity on this concept and must be attended to. In addition, I would like to ask the authors to incorporate Table 2 into Table 1: they should add a column on the key themes (which are currently reported in Table 2) and another column on the study objectives. The reviewers raised other concerns that I invite the authors to consider in their revisions.

Sincerely,

Academic Editor

Reviewers' comments:

Reviewer's Responses to Questions

**Comments to the Author**

1. Does this manuscript meet PLOS Global Public Health’s publication criteria? Is the manuscript technically sound, and do the data support the conclusions? The manuscript must describe methodologically and ethically rigorous research with conclusions that are appropriately drawn based on the data presented.

Reviewer #1: Yes

Reviewer #2: Yes

Reviewer #3: Partly

2. Has the statistical analysis been performed appropriately and rigorously?

Reviewer #1: N/A

Reviewer #2: N/A

Reviewer #3: N/A

3. Have the authors made all data underlying the findings in their manuscript fully available (please refer to the Data Availability Statement at the start of the manuscript PDF file)?

Reviewer #1: Yes

Reviewer #2: Yes

Reviewer #3: No

4. Is the manuscript presented in an intelligible fashion and written in standard English?

Reviewer #1: Yes

Reviewer #2: Yes

Reviewer #3: Yes

5. Review Comments to the Author

Reviewer #1: Dear Ms Howard and Dr Bartholomew,

I am pleased to recommend your manuscript for publication with minor revisions. Physical activity among African-American women is a very important topic given the burden of heart disease and diabetes among this population. I particularly liked the use of intersectionality as a framework and the structure of the paper. My minor suggestions are as follows:

1) You have cited each article for where the quotes have been used to explain the themes. It would be helpful if you were able to extract the data to provide age and location of the respondents. Relatedly, it would be good to know whether there were any differences in views about PA among Black women from the South versus the coasts where the influence of dominant White culture might change belief systems.

2) PLOS Global Public Health is an international journal with an international readership. Hence including a little bit of details about residential segregation and White flight might be helpful especially in the part where you discuss access. This particular theme seems very important to me but is not fully fleshed out in your paper. For example, it is helpful to know in the studies you examined, what kind of neighbourhoods and cities/towns did they focus on? Drawing from the scholarship on critical geography to explain the impacts of the external environment on the access issue for PA particularly for African-American families would strengthen the paper.

3)You have presented a table at the end of the paper which details the themes discussed in the papers. This table would benefit from including information regarding the method of the study as well as the location of the study.

4)There is no discussion of Black sports icons despite their strong presence. Was this something that the studies did not ask? And if so you need to mention that as a limitation of the reviewed studies. It would seem to me that both male and female athletes could provide good role models for PA.

Reviewer #2: This is such an important topic and lens for exploring the issue. I really enjoyed fresh approach, perspective and findings. A few minor additions could further strengthen the clarity of the analysis and implications which are so important.

A couple of ideas for how to do this: It would be helpful to briefly reference upfront how you are defining an intersectionality framework as it pertains to this analysis. This could either be a definition, basic citation of an author or conceptual model you are using or a small conceptual framework to show what you mean. For example where you mention in the discussion line 489"Of the remaining studies, only four reported themes specific to the intersectionality of Black women" which do you consider to be themes specific to intersectionality? Or could you somehow indicate which four studies those are from your table on pages 30-33 so there is greater transparency in the analysis. If for example, an intersectionality framework analyses systems of power and understanding how this manifests (eg sexism and racism) for black women in the context of exercise, body image, etc then it seems more than four of the studies and themes (noted on pages 30-33)would be considered in terms of the actual content. Or did only 4 of the studies include an analysis that drew these connections to their data? From your table I can imagine two issues - 1)the actual data collected (not asking the right questions, eg. following up on issue black women experience in terms of stereotypes, ideals of beauty or health etc. or access to places to exercise etc and 2)the researchers' understanding and labeling of themes based on the data (for example considering Doldren and Webb use of "laziness" - would you or another researcher identify that same theme in their research, or one that might have to do with time demands or competing wellness demands like connecting with and being there for family). You implicitly make this point but it would be good to present some of this more directly and explicitly.

It looks like you have a big age spread in the sample and it would be helpful to understand any major differences. Perhaps that should be listed as a potential limitation or possible avenue of future exploration.

Thank you for the opportunity to review this important topic.

Reviewer #3: Abstract

The objective is very long, complex and incomprehensible.

The technique used to examine the corpus of text from the articles is not indicated. The sentence presenting the different categories of themes identified is very long and complex.

The author should simplify and clarify the objective of the study. Long sentences should be broken down into two or three or four simple, clear sentences.

1. Introduction

The author's demonstration does not follow a guideline. I note that the author plans to examine the barriers and facilitators to physical activity among black American women. The author should propose a clear plan for reflection and demonstration. It should start with black American women and present their behavioural profiles, including their lack of physical activity. He should demonstrate that the lack of physical activity is attributable to particular socio-cultural and psychological conceptions and behaviours relating to the internalized body profile. These elements could constitute barriers to the construction of a Western or white body profile, or facilitate their systematic rejection. It could then identify a scientific problem or question based on the shortcomings probably observed in previous work on the study of barriers and facilitators linked to physical activity.The objective of the study is not clearly defined. I note several action verbs in the formulation of the objective of the study. The author should restate the objective using a single action verb.

2. Methods

The textual data analysis technique is not appropriate.

The author should use content analysis to examine the textual corpus constituted by the set of articles selected.

3. Results

The results should be presented in well-structured and structured tables.

4. Discussion

This section should be included in the fundamental lines. The author should discuss the internal validity of the methodologies used in the articles and the external validity of these methodologies.

The author should discuss the originality of the answer to the research question and the inadequacies of the systematic review.

6. PLOS authors have the option to publish the peer review history of their article (what does this mean?). If published, this will include your full peer review and any attached files.

**Do you want your identity to be public for this peer review?** For information about this choice, including consent withdrawal, please see our Privacy Policy.

Reviewer #1: **Yes: **Sreeparna Chattopadhyay

Reviewer #2: No

Reviewer #3: No

---

## [Editor Report · Decision Letter 1]

30 Oct 2024

Barriers and facilitators to physical activity among Black women: a qualitative systematic review and thematic synthesis

PGPH-D-24-00822R1

Dear Miss Howard,

We are pleased to inform you that your manuscript 'Barriers and facilitators to physical activity among Black women: a qualitative systematic review and thematic synthesis' has been provisionally accepted for publication in PLOS Global Public Health.

Best regards,

Tsitsi B. Masvawure, Ph.D.

Academic Editor

Thank you for your detailed responses to the reviewers' comments. I believe that you responded satisfactorily to the concerns raised.